# Two opposing gene expression patterns within *ATRX* aberrant neuroblastoma

**Michael R. van Gerven**[1], **Linda Schild**[1], **Jennemiek van Arkel**[1], **Bianca Koopmans**[1], **Luuk A. Broeils**[1], **Loes A. M. Meijs**[1], **Romy van Oosterhout**[1], **Max M. van Noesel**[1,2], **Jan Koster**[3], **Sander R. van Hooff**[1], **Jan J. Molenaar**[1,4]*, **Marlinde L. van den Boogaard**[1]*

1 Princess Máxima Center for Pediatric Oncology, Utrecht, Utrecht, The Netherlands, 2 Department of Cancer and Imaging, University Medical Center Utrecht, Utrecht, Utrecht, The Netherlands, 3 Department of Oncogenomics, University Medical Center Amsterdam, Amsterdam, North-Holland, The Netherlands, 4 Department of Pharmaceutical Sciences, Faculty of Science, Utrecht University, Utrecht, Utrecht, The Netherlands

* T.L.vandenBoogaard@prinsesmaximacentrum.nl (MLB); J.J.Molenaar@prinsesmaximacentrum.nl (JJM)

## Abstract

Neuroblastoma is the most common extracranial solid tumor in children. A subgroup of high-risk patients is characterized by aberrations in the chromatin remodeller ATRX that is encoded by 35 exons. In contrast to other pediatric cancer where *ATRX* point mutations are most frequent, multi-exon deletions (MEDs) are the most frequent type of *ATRX* aberrations in neuroblastoma. 75% of these MEDs are predicted to produce in-frame fusion proteins, suggesting a potential gain-of-function effect compared to nonsense mutations. For neuro-blastoma there are only a few patient-derived *ATRX* aberrant models. Therefore, we created isogenic *ATRX* aberrant models using CRISPR-Cas9 in several neuroblastoma cell lines and one tumoroid and performed total RNA-sequencing on these and the patient-derived models. Gene set enrichment analysis (GSEA) showed decreased expression of genes related to both ribosome biogenesis and several metabolic processes in our isogenic *ATRX* exon 2–10 MED model systems, the patient-derived MED models and in tumor data containing two patients with an *ATRX* exon 2–10 MED. In sharp contrast, these same processes showed an increased expression in our isogenic *ATRX* knock-out and exon 2–13 MED models. Our validations confirmed a role of ATRX in the regulation of ribosome homeostasis. The two distinct molecular expression patterns within *ATRX* aberrant neuroblastomas that we identified imply that there might be a need for distinct treatment regimens.

## Introduction

Neuroblastoma is the most common extracranial solid tumor in pediatric cancer and arises during the development of fetal adrenal neuroblasts [1, 2]. In recent years, the survival rates have improved to 81% for neuroblastoma in general [3]. However, the survival rate for high-risk patients remains only 50% despite intensive treatment. Recent data have shown that high-risk patients can be stratified in four genetic subgroups: *MYCN* amplified, *TERT* rearranged, *ATRX* aberrant or none of these three aberrations, in which *MYCN* amplifications and *ATRX*

---

**Data Availability Statement:** Data for all the isogenic models and for the tumoroid AMC772T2 have been deposited in GEO under accession number GSE226770, while the data of the cell lines

is available at ENA under accession number PRJEB55331. For the RNA-sequencing data of the iTHER data set see Langenberg et al 2022.

**Funding:** The research in this paper was supported by funding from the European Research Council (ERC) under the European Union's Horizon 2020 research and innovation programme under grant agreement numbers 716079 (Predict) and 826121 (iPC project). The funders had no role in the study design, data collection and analysis, decision to publish, or preparation of the manuscript.

**Competing interests:** The authors declare that they have no conflict of interest.

aberrations are mutually exclusive [4]. *ATRX* is a commonly mutated gene in pediatric cancer and its precise molecular role in neuroblastoma development is still unclear.

The chromatin remodeler ATRX is encoded by 35 exons localized on the X-chromosome and is involved in a plethora of nuclear processes. Its most prominent role is the chromatin incorporation of the histone variant H3.3 together with its binding partner DAXX to maintain genomic integrity and a heterochromatin state at pericentromeric and telomeric regions [5–7]. *ATRX* aberrations are associated with Alternative Lengthening of Telomeres (ALT) [8, 9]. ALT is a telomerase-independent telomeric maintenance mechanism. This mechanism is a homologous recombination-based process that is still poorly understood. A currently untested theory is that the deposition of H3.3 at telomeric regions is necessary to prevent the formation of G-quadruplexes to limit the amount of fork collapse and concomitantly double-stranded DNA breaks (DSBs) [10]. It is suggested that the process of ALT occurs due to the faulty and altered repair of these induced DSBs. Many tumors displaying ALT have *ATRX* aberrations, but several *ATRX* wild-type tumors also display ALT and how ATRX aberrations contribute to ALT is currently unknown [11]. Thus, the precise role of ATRX within the development of ALT has not yet been elucidated.

Previously, we have shown that *ATRX* multi-exon deletions (MEDs) are almost exclusively present in neuroblastoma whereas other pediatric cancers are dominated by point mutations [8]. 75% of these MEDs are predicted to be in-frame and for the most common MEDs it has been shown that these still result in protein production [8, 12], suggesting a potential gain-of-function effect compared to wild-type. So far 30 unique *ATRX* MEDs have been reported in neuroblastoma, of which only three constitute the far majority of cases [8]. Many of these 30 unique MEDs lack a large part of the N-terminal region including exons 8–9. Furthermore, several rare MEDs were discovered that lack a smaller part of the gene including exons 11–12, which contains the DAXX-binding domain. In neuroblastoma very few patient-derived models exist and the deletions that are present only represent a small fraction of those that are reported in patients, while for nonsense and missense mutations there are no model systems at all. Nonsense, missense and distinct *ATRX* deletions could be molecularly very different and therefore might need distinct therapies.

In order to study the molecular role of different *ATRX* aberrations in neuroblastoma, we created isogenic ATRX knock-out (KO) and several distinct in-frame MEDs, including some rare deletions, in neuroblastoma cell line and tumoroid models. Gene expression analysis was conducted for all generated models and for three patient-derived *ATRX* MED models. We identified two opposing molecular expression profiles for different *ATRX* aberrations.

## Results

### Characterization of neuroblastoma cell lines with *ATRX* multi-exon deletions

To study the role of ATRX aberrations in neuroblastoma development, we acquired two classical neuroblastoma cell lines, SK-N-MM and CHLA-90, and the tumoroid AMC772T2 that we had previously established in our lab [4]. We first validated the genomic aberrations of these three *ATRX* MED models using a PCR-based assay on cDNA, since the exact DNA breakpoint within the introns are unknown. For the male cell line CHLA-90 we confirmed a genomic deletion of exon 3–9 (Fig 1A), as previously reported [12, 13]. For the female cell line SK-N-MM we confirmed the nonsense mutation (Fig 1A; K1367*) as described by Qadeer et al [12], and we detected transcripts containing exon 2–9 and exon 2–10 MEDs (Fig 1A). This indicates that on the genomic level there is a deletion of exons 2–9, which is predicted to be out-of-frame. However, by skipping exon 10, in-frame transcripts are generated [14]. For the

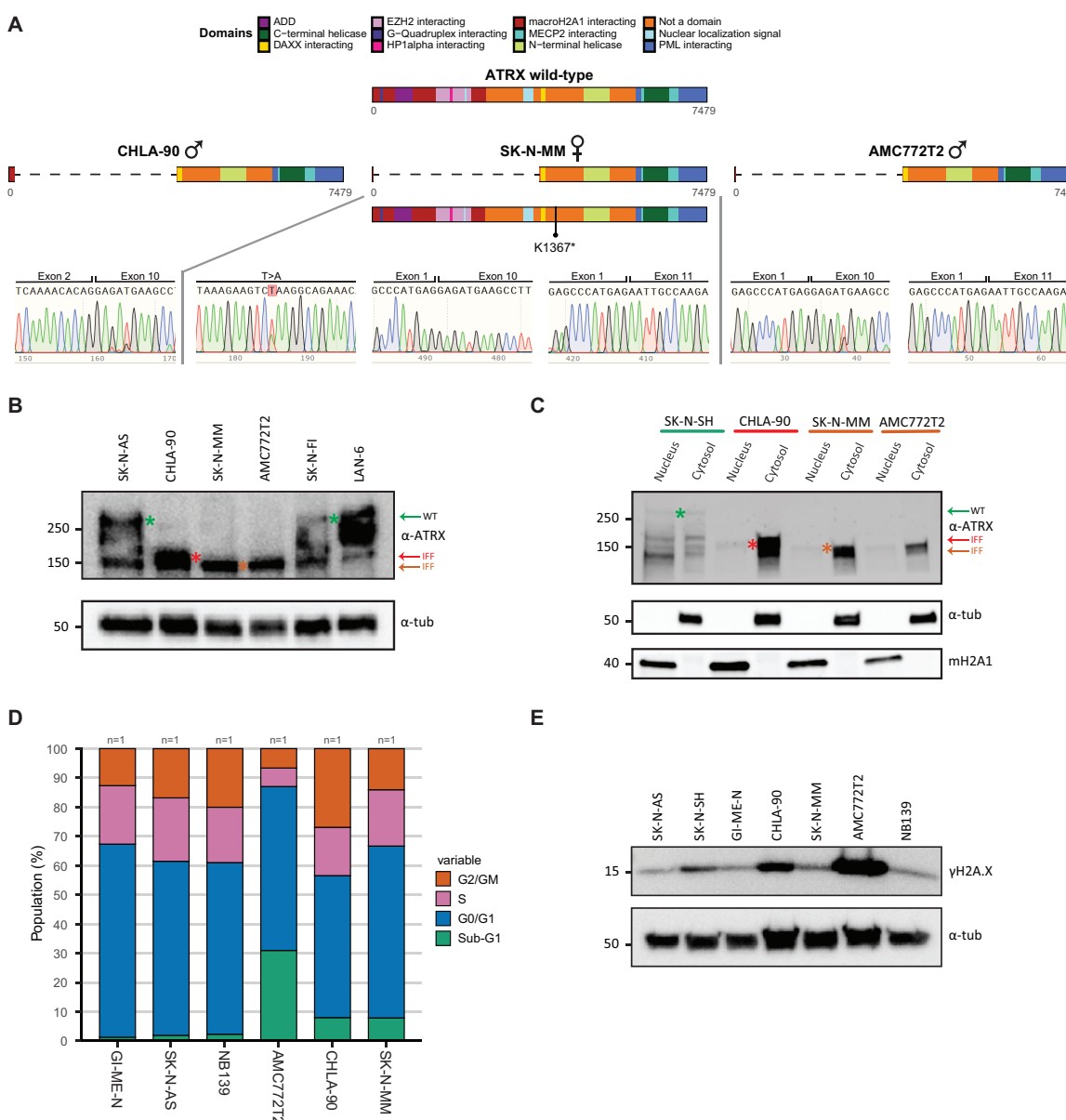

**Fig 1. Characterization of neuroblastoma cell lines with ATRX multi-exon deletions.** (a) Genetic confirmation of the *ATRX* aberrations of three patient-derived *ATRX* MED neuroblastoma models (CHLA-90, SK-N-MM and AMC772T2) utilising *ATRX*-targeted cDNA PCR amplification and sequencing. Only for the SK-N-MM point mutation we performed validation in gDNA. (b) Western blot confirming the presence of mutant in-frame fusion (IFF) ATRX protein product in the three patient-derived *ATRX* MED models. Green arrow and asterisk indicate full-length wild-type ATRX product, while red and dark orange indicate the *ATRX* exon 3–9 and 2–9 MED IFF product, respectively. Staining against α-tubulin was used as reference. (c) Western blot fractionation experiment showing cytosolic retention of the mutant IFF products. Green arrow and asterisk indicate full-length wild-type ATRX product, while red and dark orange indicate the *ATRX* exon 3–9 and 2–9 MED IFF product, respectively. The color bars on top correspond to the *ATRX* status (green: wild-type, red: exon 3–9 MED, and dark orange: exon 2–9 MED). Staining against α-tubulin and macroH2A1 were used as cytosolic and nuclear references, respectively. (d) Cell cycle distribution analysis of *ATRX* wild-type and *ATRX* MED models. "n =" indicates the number of biological replicates. For each biological replicate three technical replicates were conducted. (e) Western blot of γH2A.X abundance in *ATRX* wild-type and *ATRX* MED models. Staining against α-tubulin was used as reference.

tumoroid AMC772T2 that is derived from a male patient we found both exon 2–9 and exon 2–10 MED transcripts (Fig 1A), indicating an exon 2–9 MED on the genomic level. Thus, our patient-derived models contain *ATRX* MEDs on the genomic and transcriptomic level.

Subsequently, we assessed the protein expression of ATRX utilizing an antibody against the C-terminus that recognizes both full-length and in-frame fusion (IFF) ATRX protein products. We detected full-length ATRX protein (280 kDa; wild-type isoforms: ~250, 200 and ~150 kDa [15]) for SK-N-AS, LAN-6, and SK-N-FI (control cell lines) and ATRX IFF protein (~140–150 kDa) for CHLA-90, SK-N-MM and AMC772T2 (Fig 1B). In all three *ATRX* models, multiple key proteins domains are lost, including the nuclear localization signal (Fig 1A). Therefore, we assessed protein localization using fractionation western blot experiments, in which we detected a strong retention of IFF proteins in the cytosol with minimal amounts in the nucleus (Fig 1C). In contrast, in the ATRX wild-type cell line SK-N-SH we observed equal amounts of full-length protein in both fractions. Thus, ATRX MEDs are expressed on the protein level, but are strongly retained in the cytosol.

ATRX aberrations are strongly associated with ALT [8, 16] and therefore we employed two assays to confirm this ALT phenotype in our patient derived (PD)$^{\Delta\Delta ATRX}$ models. Our ALT-associated PML bodies (APBs) staining confirmed the presence of ALT in all three PD$^{\Delta\Delta ATRX}$ models, in which we observed strong telomeric staining that co-localized with PML protein (S1 Fig). On the telomeric southern blot we observed extremely long and heterogeneous telomeric length in our PD$^{\Delta\Delta ATRX}$ models as well as in two additional ALT models (SK-N-FI and LAN-6, both *ATRX* wild-type), but not in SK-N-SH (S2A Fig). In conclusion, we confirmed the presence of ALT in our PD$^{\Delta\Delta ATRX}$ models.

ATRX is involved in prometaphase to metaphase transition [17] and in the removal of G-quadruplexes and R-loops [18, 19]. These secondary DNA structures hinder progression of replication, which might lead to replication-fork stalling and ultimately in fork collapse and increased DNA damage. Knock-out of *ATRX* has been shown to result in both prolonged mitosis [17] and S-phase [20], the latter as a result of increased replication stress. However, we did not observe a significant increased proportion of cells in S (Padj = 0.15992) or G2/M (Padj = 1.0) phase in our PD$^{\Delta\Delta ATRX}$ models compared to wild-type models (Fig 1D, Mann-Whitney U test and Benjamini-hochberg correction). We also did not observe changes in the rate of proliferation compared to wild-type cells (S2B Fig). In contrast we observed a significant increase in the proportion of cells in sub-G1 (Padj = 0.00032908), while the proportion of G0/G1(Padj = 0.7612) were comparable (Fig 1D). Previously, it has been reported that *ATRX* knock-out leads to increased DSBs [21]. However, we only found increased DNA damage in two of the three PD$^{\Delta\Delta ATRX}$ models (Fig 1E). Thus, we detected ATRX or ALT-specific disturbances in cell cycle progression and we found no clear association between *ATRX* aberrations and the level of DNA damage.

## Generation and validation of *ATRX* aberrant isogenic cell lines/tumoroids

Currently, only few *ATRX* aberrant models are available, and they only represent a small fraction of all observed *ATRX* patient aberrations. Therefore, we made isogenic model systems, which have the additional advantage of the presence of a reference mother-line. To create *ATRX* KO models that represent *ATRX* nonsense mutations, we used a CRISPR guide targeting exon 4 and a plasmid containing homology-arms to knock-in a GFP puromycin construct for selection (Fig 2A). After selection, cells were sorted to generate single-cell clones. To generate large MEDs we used two CRISPR guides targeting the flanking intronic regions (Fig 2B) and generated single-cell clones by sorting on GFP-positive cells. We generated the most common MED of exon 2–10, the rare MED of exon 2–13 and the rare MED of exon 10–12 (for

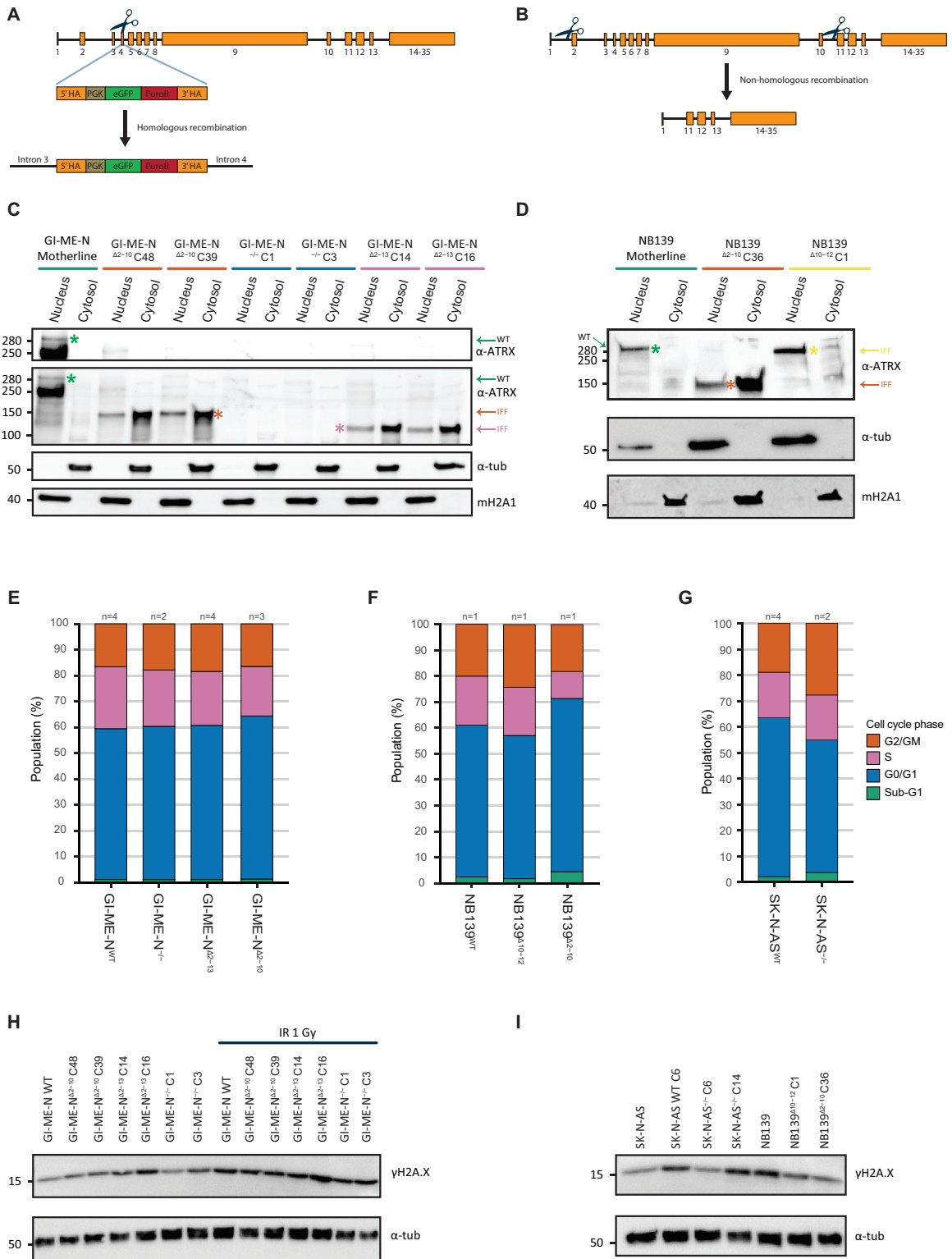

**Fig 2. Generation and validation of ATRX aberrant isogenic cell lines/tumoroids.** (a) Overview of the strategy utilised to generate ATRX KO models with CRISPR-Cas9 targeting exon 4. HA: homology arms, PGK: promotor, eGFP: green fluorescent protein, PuroR: puromycin resistance gene and scissors: guide+Cas9 (cutting region). (b) Overview of the strategy applied to create ATRX MED models with a dual-guide CRISPR-Cas9 strategy. (c-d) Western blot fractionation experiment validating cytosolic retention of the mutant *ATRX* IFF products generated by multiple isogenic (c) GI-ME-N or (d) NB139 clones. Green arrows and asterisks indicate full-length wild-type

ATRX products, while dark orange, pink and yellow indicate the *ATRX* exon 2–10, exon 2–13 and exon 10–12 MED IFF product, respectively. The color bars on top correspond to the *ATRX* status (green: wild-type, dark orange: exon 2–10 MED, blue: knock-out, pink: exon 2–13 MED and yellow: exon 10–12 MED). Staining against α-tubulin and macroH2A1 were used as cytosolic and nuclear references, respectively. (e) Cell cycle distribution analysis of *ATRX* isogenic GI-ME-N models. (f) Cell cycle distribution analysis of *ATRX* isogenic NB139 models. (g) Cell cycle distribution analysis of *ATRX* isogenic SK-N-AS models. (e-g) The number of biological replicates used in these experiments is indicated. For each biological replicate three technical replicates were conducted. (h) Western blot of γH2A.X abundance in isogenic *ATRX* aberrant GI-ME-N models at baseline and upon 1 gray irradiation (IR 1 Gy). (i) Western blot of γH2A.X abundance in isogenic *ATRX* aberrant SK-N-AS and NB139 models compared to wild-type mother-lines and clone. (h-i) Staining against α-tubulin was used as reference.

more information about all observed MED in literature see [8]). We attempted to create both *ATRX* KO and *ATRX* MEDs in several cell lines; an overview of our (un)successful attempts can be found in S1 Table and protein confirmation is shown in S3A Fig. We successfully established a total of 20 *ATRX* aberrant isogenic clones.

To further validate the isogenic models, we performed the same experiments as on our PD$^{\Delta\Delta ATRX}$ models. For all our isogenic MED models of exons 2–10 and 2–13 we detected cytosolic retention of ATRX IFF proteins (Fig 2C and 2D), similarly to what we observed in the PD$^{\Delta\Delta ATRX}$ models. Only for our isogenic MED model of exon 10–12 we observed a pattern similar to wild-type ATRX proteins (Fig 2D). In contrast to the PD$^{\Delta\Delta ATRX}$ models, we did not detect any signs of ALT in our isogenic models by APBs stainings (S4–S6 Figs) and telomeric southern blot analysis (S3B Fig). This could indicate that ALT is dependent on a more complex genomic background, which is in line with earlier reports showing that *ATRX* aberrations do not necessarily result in ALT activation [20, 22–24]. In contrast to our PD$^{\Delta\Delta ATRX}$ models, we did not observe any significant changes in cell cycle for all our isogenic model systems (Fig 2E–2G, Mann-Whitney U test and Benjamini-hochberg correction). We also observed unchanged proliferation rates for all *ATRX* aberrant GI-ME-N and SK-N-AS models (S7A and S7C Fig), while for the NB139 *ATRX* MED of exon 2–10 we observed more cells with a stronger violet trace signal compared to wild-type, indicating a decreased rate of cell proliferation (S7B Fig). Lastly, we assessed γH2A.X levels as a measure of the amount of DSBs in our isogenic models but did not observe increased DNA damage (Fig 2G and 2H), not even upon induction by irradiation in our GI-ME-N models (Fig 2G). In summary, our created isogenic models recapitulated some of the phenotypes observed in the PD$^{\Delta\Delta ATRX}$ models.

## Strong overlap of differentially expressed genes between *ATRX*$^{\Delta 2–13}$ and *ATRX*$^{-/-}$ GI-ME-N models

To study the molecular landscape of *ATRX* aberrant neuroblastoma we assessed the transcriptomes by performing total RNA-sequencing. The principal component analysis (PCA) on all isogenic models showed a separation based on the mother-lines (S8A Fig). From the PCA of GI-ME-N, SK-N-AS and NB139 we observed a clear separation of wild-type versus *ATRX* mutant clones (S8B–S8D Fig). However, for the GI-ME-N wild-type clones we observed separation between the clones on PC1 (S8B Fig), suggesting the presence of a batch effect. This batch effect can be attributed to the two batches in which we generated the isogenic GI-ME-N models (as wild-type clones 3 and 4 were generated together with the *ATRX* exon 2–10 MED models). Therefore, we performed our differential expression analysis for the distinct GI-ME-N *ATRX* aberrant models with the corresponding wild-type clones. Lastly, for an unknown reason we observed clustering of GI-ME-N$^{\Delta 2–13}$ (*ATRX* MED exon 2–13) clone 15 with the GI-ME-N$^{-/-}$ clones. However, as this is only observed for a single clone it should have negligible impact on the analysis.

We performed differential expression analyses for SK-N-AS$^{-/-}$ (*ATRX* KO), GI-ME-N$^{-/-}$, GI-ME-N$^{\Delta2-10}$ (*ATRX* MED exon 2–10), GI-ME-N$^{\Delta2-13}$ and NB139$^{\Delta2-10}$ by comparing the *ATRX* aberrant clones to the corresponding wild-type clones (Fig 3A). Several thousands of genes were differentially expressed in each of the five expression analyses (S2 Table). To determine if the different *ATRX* aberrations resulted in similar changes in gene expression, we took the overlap of both the down- and upregulated genes within the GI-ME-N models. We noticed a striking overlap between GI-ME-N$^{-/-}$ and GI-ME-N$^{\Delta2-13}$ and very little overlap with GI-ME-N$^{\Delta2-10}$, indicating that there might be two distinct expression patterns within *ATRX* aberrant GI-ME-N models (Fig 3B). Therefore, this suggests one expression pattern related to complete inactivation of ATRX (GI-ME-N$^{-/-}$ and GI-ME-N$^{\Delta2-13}$) and one related to the IFF with remaining or gained protein activity (GI-ME-N$^{\Delta2-10}$).

## Gene ontology reveals increased expression of genes related to metabolic process in *ATRX*$^{\Delta2-13}$ and *ATRX*$^{-/-}$ models and decreased expression in *ATRX*$^{\Delta2-10}$ models

Our above analyses suggested one expression pattern related to complete *ATRX* inactivation (*ATRX*$^{-/-}$ and *ATRX*$^{\Delta2-13}$) and one related to the IFF (*ATRX*$^{\Delta2-10}$). To test this hypothesis, we looked at the overlapping down- and upregulated genes between all KO and *ATRX*$^{\Delta2-13}$ models and performed gene ontology (GO) analysis on those genes to identify which genes are always differentially expressed as a result of the *ATRX* aberrations, irrespective of the (epi) genetic background of the distinct cell line or tumoroid models. GO analysis of the 227 overlapping downregulated genes showed enrichment for the regulation of several RNA and metabolic processes (Fig 3C), while for the upregulated genes we found an overlap of 142 genes that are enriched for seven processes involved in metabolism (Fig 3D). In contrast, GO analysis of the 618 overlapping downregulated genes for our *ATRX*$^{\Delta2-10}$ models identified these same seven metabolic processes and several other terms involved in other metabolic and RNA processes (Fig 3E and S9 Fig). GO analysis of the 635 overlapping upregulated genes for our *ATRX*$^{\Delta2-10}$ models showed enrichment for process involved in biological quality and localization (Fig 3F). In all four Venn diagrams (Fig 3C–3F) we also noticed many non-overlapping genes that were only differentially expressed in a single cell line model. This might be explained by the fact that ATRX is a chromatin remodeler and that the effects of *ATRX* aberrations are highly dependent on the epigenetic landscape that is present in the distinct cell lines.

ATRX is known to bind to the 3' exon of zinc finger genes [25] where it could potentially modulate their expression. Therefore, we performed Panther protein class analysis [26] for the overlapping up and down-regulated genes for both the *ATRX*$^{-/-}$ and *ATRX*$^{\Delta2-13}$ models and for the *ATRX*$^{\Delta2-10}$ models. Only for the downregulated genes of both we found significant terms, namely zinc finger transcription factors for our *ATRX*$^{-/-}$ and *ATRX*$^{\Delta2-13}$ models (S8G Fig) and proteins involved in RNA processes and translation in *ATRX*$^{\Delta2-10}$ models (S8H Fig). Altogether, two distinct expression profiles are present within *ATRX* aberrant neuroblastoma that seem to lead to opposing changes in metabolic processes.

## GSEA reveals two opposing expression patterns within *ATRX* aberrant neuroblastomas

To acquire more understanding of the changed biological processes we performed Gene Set Enrichment Analysis (GSEA) for all isogenic model systems. Additionally, we performed GSEA for the comparison of PD$^{\Delta\text{ATRX}}$ models with five non-*MYCN* amplified neuroblastoma cell lines (S8E Fig, for number of DEGs see S2 Table) and for the comparison of neuroblastoma tumors from the individualized THERapy (iTHER) project. The iTHER data contains 2

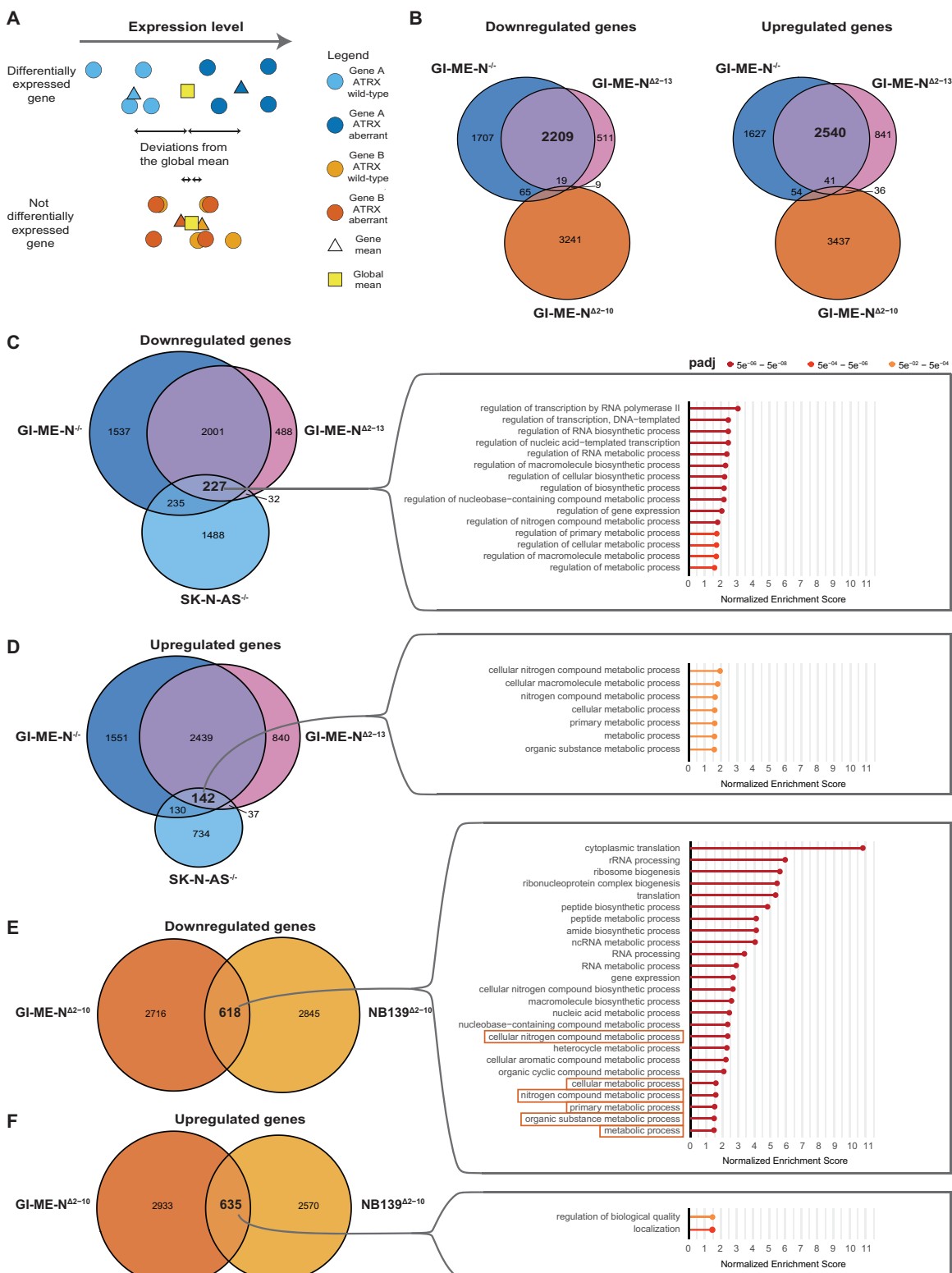

**Fig 3. Overlap of differentially expressed genes (DEGs) and gene ontology (GO) analysis identifies two distinct gene expression patterns within *ATRX* aberrant neuroblastoma.** (a) Simplified overview of differential expression analysis for two genes for *ATRX* wild-type and aberrant models. (b) Overlapping DEGs for both the down and upregulated genes within the isogenic *ATRX* aberrant GI-ME-N models. (c) GO analysis of the overlapping downregulated DEGs between all *ATRX*[-/-] and *ATRX*[Δ2–13] isogenic models. (d) GO

analysis of the overlapping upregulated DEGs between all *ATRX*$^{-/-}$ and *ATRX*$^{\Delta2-13}$ isogenic models. (e) GO analysis of the overlapping downregulated DEGs between both *ATRX*$^{\Delta2-10}$ isogenic models. Only the top 25 significant gene ontologies are shown here; for all significant GO terms, see S9 Fig. Orange boxes highlighted the same terms as observed in Fig 3D. (f) GO analysis of the overlapping upregulated DEGs between both *ATRX*$^{\Delta2-10}$ isogenic models.

tumors with an *ATRX* MED of exon 2–10 (iTHER$^{\Delta2-10}$) and 7 *ATRX* wild-type and non-*MYCN* amplified tumors (S8F Fig, for number of DEGs see S2 Table). Thus, in total we per-formed 7 GSEA for both the gene set database gene ontology biological process (GO BP) and Reactome (Fig 4A). The most common *ATRX* aberration in neuroblastoma is a MED of exon 2–10 [8] and therefore we visualized all the overlapping significantly changed genes sets with the same Normalised Enrichment Score (NES) directionality (i.e.–or +) between GI-ME-N$^{\Delta2-10}$, NB139$^{\Delta2-10}$, PD$^{\Delta\Delta ATRX}$ and iTHER$^{\Delta2-10}$ in bubble plots (Fig 4A). The bubble plot of all the overlapping significant GO BP gene sets shows decreased expression of genes related to ribo-some biogenesis, translation, and metabolism in these models (Fig 4B, S10A–S10D Fig). In sharp contrast we observed the complete opposite patterns for GI-ME-N$^{-/-}$, SK-N-AS$^{-/-}$ and GI-ME-N$^{\Delta2-13}$, which is in line with our GO analysis results (Fig 4B, S10E–S10G Fig). This pattern was confirmed by the comparisons of the GSEA of Reactome (Fig 4C). Taken together, this suggests two opposing expression profiles within *ATRX* aberrant neuroblastoma.

McDowell et al. [7] reported that ATRX binds to the short arm of acrocentric chromosomes where rDNA copies are localised. Together with our GSEA data this suggests a role of ATRX in ribosomal biogenesis. Interestingly, we also observed multiple GO BP gene sets in our GSEA related to cytoplasmic translation, mitochondrial translation, and metabolism (Fig 4B). These GO BP gene sets are all dependent on the abundance of ribosomes, since a reduction or increase in the number of ribosomes leads to reduced or increased translation capability [27] and consecutively might lead to changed metabolism. Visualization of the top 50 differentially expressed ribosome biogenesis genes for all our isogenic models and cell line data showed changed expression of both small and large ribosomal proteins and also of many other proteins involved in ribosome biogenesis (Fig 4D–4F, S11A–S11C Fig). This suggests that IFFs in *ATRX* may lead to modulations of ribosome homeostasis.

## ATRX is involved in ribosome biogenesis by modulating rRNA expression

MYCN, c-MYC and the ATRX binding partner EZH2 are all directly involved in regulating ribosome biogenesis [28–30]. To exclude an indirect effect of ATRX on ribosome biogenesis via the expression of these genes, we assessed their protein abundance. For both MYCN and c-MYC we observed unchanged expression in the isogenic model systems, while for EZH2 we observed a slight decrease in expression in the GI-ME-N$^{\Delta2-10}$ clones and a strong decrease in the NB139$^{\Delta2-10}$ clone (S12A Fig). However, we also observed decreased EZH2 protein expres-sion in SK-N-AS$^{-/-}$ clone 14. This pattern in changed EZH2 expression is not in line with our RNA data, as we observed opposing expression patterns for ATRX$^{\Delta2-10}$ and ATRX$^{-/-}$ models. Thus, we can exclude that ATRX modulates ribosome biogenesis indirectly via expression of these three genes. Additionally, we assessed the gene expression of the *REST* gene, which was previously reported to be overexpressed in CHLA-90 and SK-N-MM compared to LAN-6 and SK-N-FI [12]. However, we did not observe overexpression in both our cell line and tumor data (S12B and S12C Fig).

As mentioned above, ATRX binds to the short arms of acrocentric chromosomes where rDNA copies are localized [7]. Therefore, it could be involved in modulating the chromatin landscape at these regions and in that manner regulate rRNA expression. We assessed the

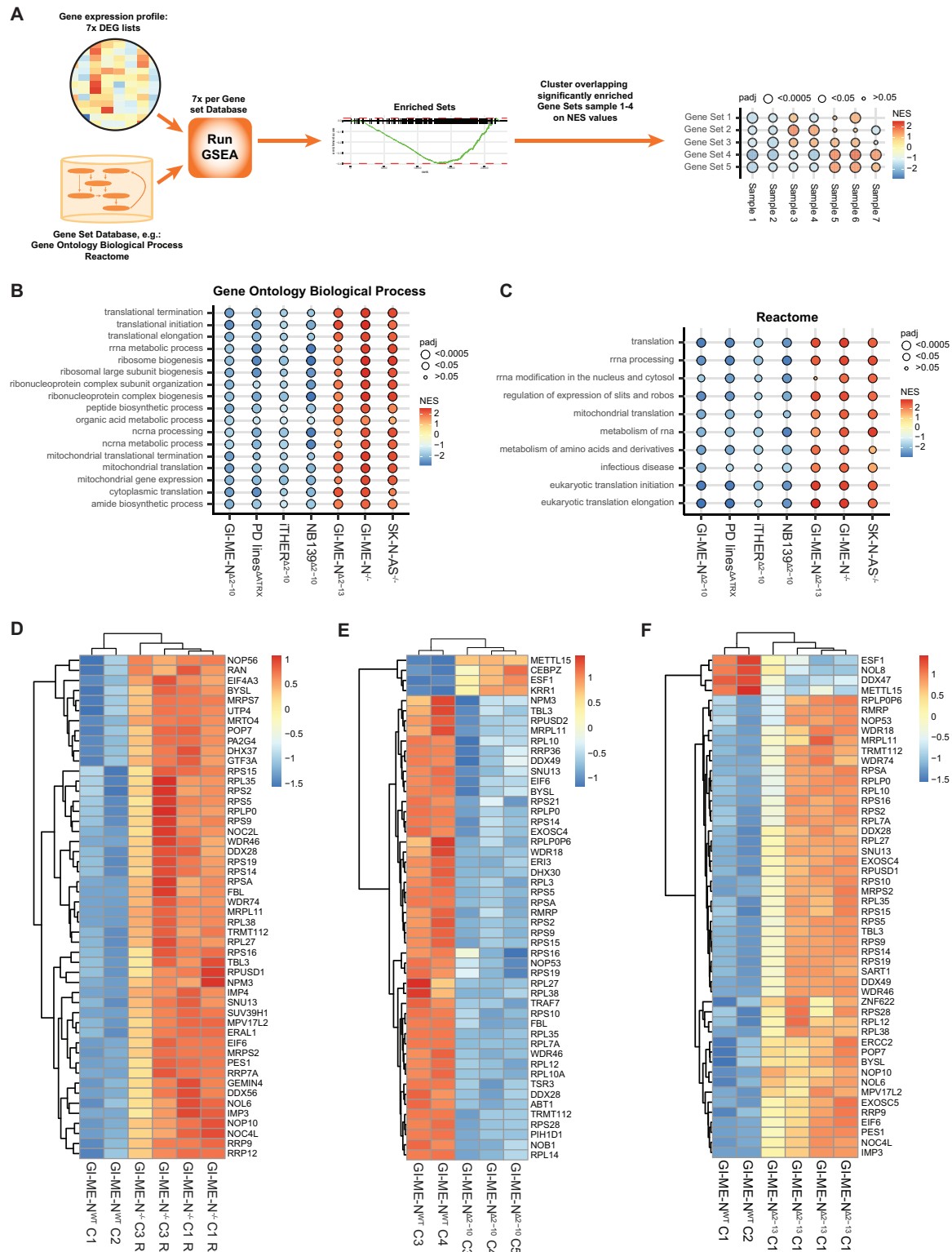

**Fig 4. Gene set enrichment analysis (GSEA) reveals two opposing expression patterns within *ATRX* aberrant neuroblastoma related to ribosome biogenesis.** (a) Overview of the strategy to combine the 7 GSEA for our 7 differentially expressed gene (DEG) lists. NES: Normalised Enrichment Scores. (b-c) Bubble plot showing all gene sets that were both significant and showed the same directionality of the NES values (positive or negative) in GI-ME-N$^{\Delta2-10}$, NB139$^{\Delta2-10}$, PD$^{\Delta ATRX}$ and iTHER$^{\Delta2-10}$ for (b) the GO BP gene set database or (c) the Reactome gene set database. (d-f) Heatmap of the top 50 differentially expressed ribosome biogenesis genes for (d)

the isogenic GI-ME-N$^{-/-}$ clones, for (e) the isogenic GI-ME-N$^{\Delta2-10}$ clones and for (f) the isogenic GI-ME-N$^{\Delta2-13}$ clones. The heatmaps display expression values that were normalized across all samples by Z-score. Both row and column clustering were applied using the Euclidean distance.

rRNA expression in our isogenic models and in the PD$^{\Delta\Delta ATRX}$ models by performing qPCRs on the unspliced 47S pre-rRNA. We observed increased rRNA expression for GI-ME-N$^{-/-}$, SK-N-AS$^{-/-}$ and GI-ME-N$^{\Delta2-13}$ (two sample t-test assuming unequal variance: p = 0.015, p = 0.0245, p = 0.177, respectively) and decreased rRNA expression in GI-ME-N$^{\Delta2-10}$, NB139$^{\Delta2-10}$ and PD$^{\Delta\Delta ATRX}$ (two sample t-test assuming unequal variance: p = 0.354, p = 0.029, p = 0.0011, respectively) models (Fig 5A–5D). These data support a potential role of *ATRX* in ribosome biogenesis, since the changes in rRNA expression we observed are in line with the differential gene expression pattern. Lastly, we also assessed the rRNA expression in two CHLA-90 clones with doxycycline inducible *ATRX* wild-type expression. These clones still expressed the IFF protein, and this expression was unchanged upon doxycycline induction (S12D Fig), while *ATRX* wild-type expression was only detectable upon induction (S12E Fig). We observed a lower, yet not statistically significant, rRNA expression upon doxycycline induction (Fig 5E; two sample t-test assuming unequal variance, p = 0.068). In conclusion, ATRX is likely involved in ribosome biogenesis and *ATRX* abrogation leads to changed rRNA expression.

## Discussion

The aim of this study was to assess whether different *ATRX* aberrations are molecularly distinct from one another and how they might contribute to tumor development. We developed a total of 20 isogenic clones of several distinct *ATRX* aberrations. Our RNA analysis revealed a strong overlap in gene expression between *ATRX*$^{\Delta2-13}$ and *ATRX*$^{-/-}$ and very little overlap with *ATRX*$^{\Delta2-10}$ models. Moreover, we found opposing expression patterns between the *ATRX*$^{\Delta2-13}$ and *ATRX*$^{-/-}$ aberrations compared to the *ATRX*$^{\Delta2-10}$ aberrations for ribosome biogenesis, non-coding RNA processes and several metabolic processes. Lastly, we showed evidence of a potential role of *ATRX* in ribosome biogenesis.

*ATRX* aberrations are strongly associated with ALT and all neuroblastomas with *ATRX* aberrations tested so far utilize this telomere maintenance mechanism [8]. However, we did not observe ALT in any of our *ATRX* aberrant isogenic model systems. The cell lines and tumoroids in which we attempted to make *ATRX* aberrations were all telomerase-dependent and according to literature telomerase-dependent telomere maintenance could be favored over ALT [31, 32]. Therefore, we also attempted to KO *TERT* or *TERC* to force cells in using ALT, but no cells survived, and this could therefore indicate that other factors might be necessary in conjunction to induce ALT. There is indeed evidence that inducing aberrations within the *ATRX* gene does not necessarily cause ALT in different cell types and cancers [20, 22–24].

ATRX has been reported to be involved in cell cycle progression, since *ATRX* KO resulted in prolonged mitosis [17] and S-phase [20], the latter as a result of increased replication stress. Interestingly, we did not observe any changes in these two cell cycle phases for both our isogenic *ATRX* aberrant and PD$^{\Delta\Delta ATRX}$ models. Potentially indicating that the effect of *ATRX* aberrations is highly dependent on the tissue of origin, as the changes in mitosis and S-phase were observed in HeLa cells [17] and mouse embryonic stem cells [20], respectively. In contrast to our isogenic *ATRX* aberrant models, we did observe a significant increase in the population of sub-G1 cells in the PD$^{\Delta\Delta ATRX}$ models. This difference might be explained by the presence of ALT in the PD$^{\Delta\Delta ATRX}$ models, as ALT cells are highly heterogenous in telomere

**A**

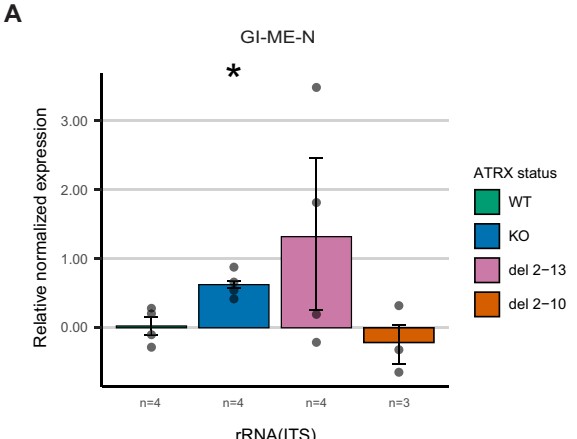

**B**

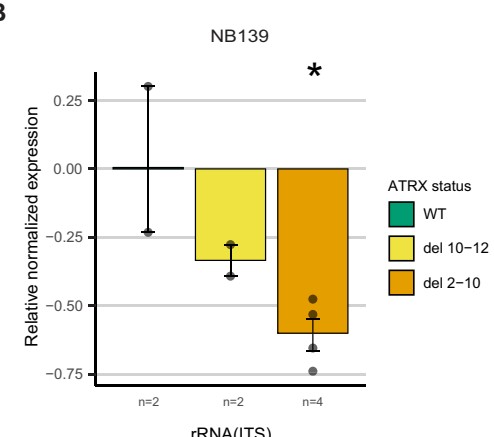

**C**

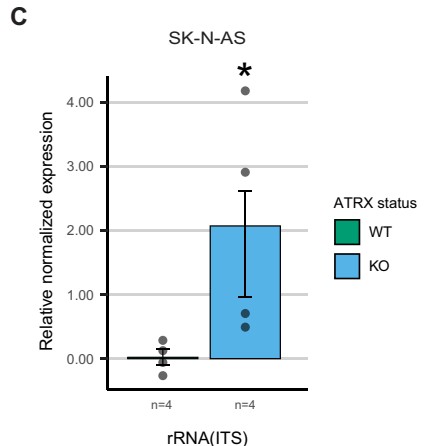

**D**

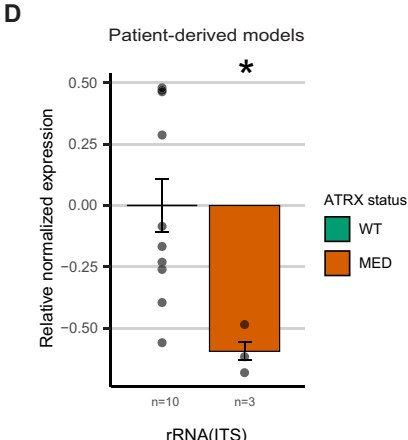

**E**

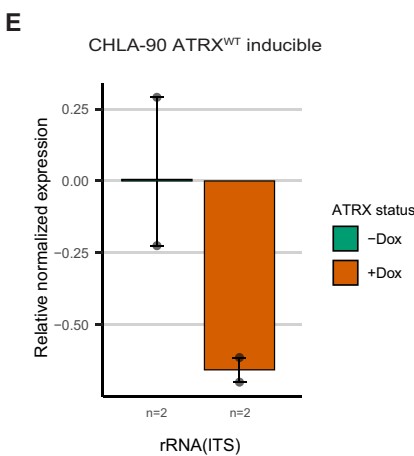

**Fig 5. ATRX is likely involved in ribosome biogenesis by modulating rRNA expression.** qPCR- validation on the unspliced 47S pre-RNA using primers for ITS for (a) the isogenic GI-ME-N clones (p-values GI-ME-N$^{-/-}$: 0.015; GI-ME-N$^{\Delta2-10}$: 0.354 and GI-ME-N$^{\Delta2-13}$: 0.177). (b) the isogenic NB139 clones (p-values NB139$^{\Delta10-12}$: 0.277 and NB139$^{\Delta2-10}$: 0.029). (c) the isogenic SK-N-AS clones (p-value: 0.0245). (d) the patient-derived *ATRX* MED models (p-value: 0.0011). (e) the isogenic *ATRX*$^{WT}$ doxycycline-inducible CHLA-90 clones (p-value: 0.068). (a-e) the number of biological replicates used is indicated below each bar. Each dot represents the average of three

technical replicates. For statistical analyses t-tests assuming unequal variance were used, and an asterisk indicates a significant difference. The whiskers represent the standard errors of the mean.

length and include many cells with critically short telomeres resulting in increased cell death and senescence [33–35]. To date only a few studies have assessed the cell cycle for *ATRX* aberrant cells and therefore more thorough analysis should be conducted in the future, preferably on *ATRX* aberrant cells originating from distinct tissues.

Changed ribosome biogenesis, proliferation and metabolic processes have been observed in many cancers [36]. In this study we identified a dichotomy in ribosome biogenesis and several related metabolic processes between the $ATRX^{\Delta2-10}$ and the $ATRX^{\Delta2-13\&-/-}$ ($ATRX^{\Delta2-13}$ and $ATRX^{-/-}$) neuroblastoma tumor cells. Additionally, we showed that rRNA expression is downregulated in $ATRX^{\Delta2-10}$ models and upregulated in $ATRX^{\Delta2-13\&-/-}$ models, completely in accordance with our gene expression data. This suggests that ATRX is involved in ribosome homeostasis either through direct or indirect modulation of ribosome biogenesis. The first hint on the involvement of ATRX in ribosome biogenesis dates from 1999, when it was discovered that ATRX binds to rDNA arrays during metaphase [7]. Only very recently it was observed that ATRX binds to several proteins directly involved in ribosome biogenesis [37]. Another report showed that ATRX binds to the promotor region of rDNA and they observed increased ribosome biogenesis in gliomas with nonsense mutations, which is in line with our generated *ATRX* KO models [38]. This could suggest that increased ribosome biogenesis as a result of an *ATRX* nonsense mutation or KO, is a more general phenomenon in cancer. Nevertheless, more research in additional tumor types is needed to confirm whether this is a universal phenomenon in *ATRX* KO cells. Intriguingly, we did not observe changed proliferative rates for the majority of our created models, but we observed several altered metabolic processes in our data. This could suggest that the changes in ribosome biogenesis did not reach the threshold to change the proliferation rate but could have reached the threshold to rewire metabolism. There are recent indications that changes in ribosome biogenesis can directly modulate metabolism [39].

In the majority of cancers, high rates of ribosome biogenesis are observed and contribute to tumorigenesis [40]. However, for our $ATRX^{\Delta2-10}$ models we observed downregulation of ribosome biogenesis. Nevertheless, decreased ribosome biogenesis has also been shown to promote tumorigenesis as is observed for patients with ribosomopathies that are prone to the development of certain tumor types [40]. It is postulated that the lower amounts of ribosomes leads to competition between various mRNAs and that tumor suppressor encoding mRNAs with lower ribosomal binding affinity could lead to reduced expression of certain tumor suppressors [40]. Also, recent evidence shows that decreases and increases of specific ribosomal proteins are advantageous for tumor development and that these effects are highly cell type and tissue specific [39]. Hopefully, further research will identify and illuminate the precise role of ATRX in ribosome biogenesis.

The distinction between the $ATRX^{\Delta2-10}$ and $ATRX^{\Delta2-13}$ models that we observed is remarkable, as the only difference is the deletion of two extra exons in the $ATRX^{\Delta2-13}$ models. None of the known binding regions reside within these exons. However, for many ATRX interaction partners the binding regions within the ATRX protein are still undetermined. A potential explanation for the discrepancy between these two models could be that the $ATRX^{\Delta2-10}$ IFF products still have a remaining function or sequester away binding partners in the cytosol compared to the $ATRX^{\Delta2-13}$ IFF products. Although we showed that most of the $ATRX^{\Delta2-10}$ and $ATRX^{\Delta2-13}$ IFF products reside in the cytosol, a small fraction is located in the nucleus. A

similar distribution pattern has been shown previously for SK-N-MM and CHLA-90 in a study by Qadeer et al. [12] In this study, they also showed that the small fraction of nuclear localized ATRX IFF products ($ATRX^{\Delta2-10}$ and $ATRX^{\Delta3-9}$ for SK-N-MM and CHLA-90, respectively) still binds chromatin, although with an alternative distribution, and that this resulted in altered gene expression. Thus, an alternative explanation for the discrepancy that we observe could be that the $ATRX^{\Delta2-13}$ IFF products are incapable to bind chromatin and therefore their effect on gene expression could be similar to *ATRX* KO. One limitation of our study is the fact that we only successfully generated the $ATRX^{\Delta2-13}$ aberrations in a single cell line system and therefore we cannot rule out the possibility that the observed expression pattern for this model may be cell line-specific. Additionally, we showed that the $ATRX^{\Delta10-12}$ IFF products are the only MED products that predominantly localize to the nucleus. In the future it might be interesting to further examine the effect of this specific MED. Currently, we also do not know whether this MED plays a role in ALT, which at this moment cannot be assessed due to lack of patient material. Hopefully, future endeavors will unravel the other factors necessary for ALT induction and enable further investigation into this small *ATRX* MED.

In this study we successfully established multiple isogenic *ATRX* aberrant models with several distinct *ATRX* aberrations. Utilizing these models, we identified two opposing expression patterns within *ATRX* aberrant neuroblastoma and a potential role of *ATRX* in ribosome biogenesis. Lastly, we want to emphasize that the observed dichotomy in expression pattern within *ATRX* aberrant neuroblastoma suggests the potential need for two distinct treatment regimens, since the $ATRX^{\Delta2-10}$ and the $ATRX^{\Delta2-13\&-/-}$ tumor cells are molecularly very distinct and are therefore likely to respond differently to the same treatment regimen.

## Materials and methods

### Cell culture

All neuroblastoma cell lines used in this study were obtained from the American Type Culture Collection or via historic collaborations. Neuroblastoma tumoroids used in this study have been established in our group as previously described [41, 42]. The neuroblastoma cell lines and tumoroids were cultured in various culture media (S3 Table). Originally, we grew NB139 in TIC medium, but at some point in time our isogenic *ATRX* aberrant NB139 models stopped growing in this medium and therefore we switched to tumoroid medium with 20% human plasma for both the *ATRX* aberrant and wild-type NB139 clones (wild-type clones had no problem growing in TIC medium; S3 Table). All cells were grown in an incubator at 37°C and 5% $CO_2$. Cell lines and tumoroids were routinely checked for mycoplasma infections and authenticated through short tandem repeat profiling.

### sgRNA design and plasmid generation

sgRNAs were designed using the CRISPOR design tool [43] (sgRNA sequences are listed in S4 Table) and were cloned into the pSpCas9(BB)-2A-GFP (PX458) (this plasmid was a gift from Feng Zhang [44], addgene plasmid #48138). The cloning of the sgRNAs and the two homology arm plasmids (one for ATRX_KO_sgRNA_1 and one for ATRX_KO_sgRNA_2) were performed as described by Boogaard et al [45]. The primers used for amplification of the homology arms and the PGK-eGFP-puromycin cassette are listed in S5 Table.

For cloning of the PiggyBac doxycycline-inducible mCMV-Kozak-ATRX-wildtype plasmid we first performed a PCR on cDNA using primers SalI_Kozak_ATRX_FW and ATRX_cDNA_RV (S5 Table). Next, we PCR amplified a mCMV from a plasmid using primers XhoI_attL1_mCMV_FW and SalI_mCMV_RV followed by T7 ligation (NEB) of both

PCR products. The resulting product was cloned into a pJET1.2/blunt vector (cloneJet PCR cloning Kit, Thermo Scientific, K1231). Subsequently, the resulting plasmid and the IF-GF-P-ATRX plasmid (a gift from Michael Dyer [46], addgene plasmid #45444; exon 6 absent) were digested with XhoI (Promega) and SpeI (Promega) and ligated to generate a Attl1-mCMV-Kozak-ATRX plasmid including exon 6. Next, PCR amplification was performed on the IF-GFP-ATRX plasmid using primers BstEII_ATRX_FW and MluI_Att2L_RV and the resulting product was cloned into a pJET1.2/blunt vector. Subsequently, the resulting plasmid and the Attl1-mCMV-Kozak-ATRX plasmid were digested with BstEII-HF (NEB) and MluI-HF (NEB) and ligated together. The resulting Attl1-mCMV-Kozak-ATRX-Attl2 plasmid was digested using KasI (NEB) and the PB-TA-C-ERN plasmid (a gift from Knut Woltjen, addgene plasmid #80475) was digested with MluI followed by a LR gateway clonase reaction (ThermoFisher, 11791020) for 18 hours. All bacterial plates for the cloning of the PiggyBac doxycycline-inducible mCMV-Kozak-ATRX-wildtype plasmid were grown at room temperature.

## Establishing isogenic models (transfections and clone selection)

*ATRX* knock-out clones were established by transfecting SK-N-AS and GI-ME-N with the Cas9 expressing vector containing ATRX_KO_sgRNA_2 (targeting exon 4) and the corresponding homology arm plasmid using Fugene HD transfection reagent (E2312, Promega). Three days after transfection, medium containing 1.5 ug and 1 ug/mL Puromycin (Sigma, P8833) was added to SK-N-AS and GI-ME-N cells respectively. Several weeks after transfections SK-N-AS cells were single-cell sorted by FACS using the SH800S (Sony Biotechnology) sorter in 96-well plates and GI-ME-N cells were single-cell diluted in 96-well plates. Clonal cultures were expanded and harvested for gDNA, protein and RNA to confirm editing.

*ATRX* MED clones were established by transfecting GI-ME-N and NB139 cells with two Cas9 expressing vectors containing two distinct sgRNAs (S4 Table) using Fugene HD transfection reagent. Three days after transfection, GFP-positive (transiently expressed) cells were single-cell sorted and expanded as described above. Genomic DNA, protein and RNA was harvested to confirm genome editing. For GI-ME-N, we expected only one *ATRX* allele (X-chromosomal loss according to WGS data), however our GI-ME-N cells contained three alleles. Presence of MEDs was confirmed on only one allele in all GI-ME-N clones. Subsequently, we transfected GI-ME-N cells with an *ATRX* exon 2–13 MED with ATRX_-KO_sgRNA_1 and the corresponding homology arm plasmid to KO the remaining wild-type alleles using Fugene HD transfection reagent. For some GI-ME-N clones with an exon 2–10 MED, we transfected them with either ATRX_KO_sgRNA_1 (clone 7, 31 and 39) or ATRX_-KO_sgRNA_2 (all other clones) and the corresponding homology arm plasmids using Fugene HD transfection reagent. Three days after transfection, medium containing 1 μg /mL Puromycin was added to the cells and several days or weeks later, cells were single-cell sorted by FACS in 96-well plates. Clonal cultures were expanded and harvested for gDNA, protein and RNA to confirm editing.

CHLA-90 doxycycline-inducible *ATRX* wild-type clones were established by transfecting cells with 50 ng PiggyBac doxycycline-inducible mCMV-Kozak-ATRX-wildtype plasmid and 50 ng of the pCMV-hyPBase plasmid (a gift from Kosuke Yusa [47], Wellcome Trust Sanger Institute) in 12-well plates. Three days after transfection, medium containing 1000 μg/mL Neomycin (G-418, Roche) was added to the cells. Several weeks later, cells were single-cell sorted by FACS in 96-well plates. Clonal cultures were expanded, and protein was harvested for cells treated with and without 2500 ng/ml doxycycline to confirm editing.

## gDNA and cDNA validation of clones

For genotyping, we extracted genomic DNA utilizing the Wizard® SV Genomic DNA Purification System (Promega). Primers were designed to amplify the allele with the PGK-eGFP--Puromycin insert, the wild-type allele or the allele containing distinct MEDs (S6 Table). For mRNA expression validations, cDNA was generated using 2.5 microgram of RNA and the IScript cDNA Synthesis Kit (1708891, Bio-Rad) according to the manufacturer's manual. Primers were designed to amplify the wild-type or the distinct MEDs allele mRNA products. PCR products were analysed by gel electrophoresis and Sanger sequencing.

## Western blot analysis

Western blots were performed as described in Boogaard et al [45]. Cell fractionation experiments were performed with the ProteoExtract® Subcellular Proteome Extraction Kit (Calbiochem®). Primary and secondary antibodies are shown in S7 Table.

## ALT southern blot analysis

DNA for southern blot was isolated in 10 ml SE buffer (75mM NaCl, 25mM Na2, EDTA, pH 8.0) and extracted using phenol-chloroform extraction. Southern blots to detect ALT were performed using the TeloTAGGG Telomere Length Assay Kit (12209136001, Sigma) according to the manufacturer's protocol.

## ALT-associated PML bodies (APBs) staining

APBs stainings were performed on cells grown on coverslips in 6-well plates. For APBs stainings, coverslips were washed twice with 1x PBS and incubated with 4% paraformaldehyde for 20 minutes. Cover glasses were incubated for 3 minutes with 70%, 2 minutes with 95% and 2 minutes with 100% ethanol. Next, coverslips were airdried and put on a coverglass with 10 μL of probe in-between (10 μL HB buffer (50% formamide, 10% Dextran Sulfate Sodium and 2x SSC (saline-sodium citrate)) and 0.5 μL TelC-Alexa-Fluor-488 PNA Bio probe (F1104)). The coverslips were fixed on the coverglass using Fixogum and airdried for 1 hour. Denaturation was performed for 2 minutes at 75˚C and the slides were incubated overnight in a wet hybridization chamber at 37˚C. The next day, the Fixogum and coverglass were removed and coverslips were washed with 2x SSC buffer for 5 minutes, followed by 1x PBS wash. Permeabilization was performed for 5 minutes using 1x PBS/0.1X triton X-100 followed by 1x PBS wash. Next, coverslips were incubated for 5 minutes with 10 mM Sodium Citrate (pH 6), after which 30 minutes blocking was performed using TBS/1% BSA/0.1% Triton X-100. Coverslips were incubated overnight at 4˚C with the primary PML antibody (See S7 Table). The next day, coverslips were washed three times with 1x PBS and incubated with the secondary antibody (S7 Table) for 2 hours at room. Subsequently, coverslips were washed 3 times with 1x PBS followed by incubation for 3 minutes with 70%, 2 minutes with 95% and 2 minutes with 100% ethanol. Lastly, coverslips were airdried, DAPI stained and sealed on a coverglass. Imaging was performed on a Leica DM RA microscope with a 63 mm lens.

## Cell cycle analysis

Cells were harvested, washed in 1x PBS and resuspended in 1x PBS 2mM EDTA until single cells were obtained. Subsequently, cells were washed with 1x PBS and 1 million cells were stained with 1x PBS containing 1:1000 Zombie NIR™ (BioLegends). Cells were incubated for 20 minutes at room temperature and washed with 1x PBS. Fixation of the cells was performed with 200 μL fixation buffer (eBioscience™ Foxp3/Transcription Factor Staining Buffer, Set,

Invitrogen^TM^) and incubated for 30 minutes at 4˚C. Next, cells were washed twice with 500 μL permeabilization buffer and washed with FACS buffer (2% FCS 2mM EDTA 1x PBS), after which the cells were stained in FACS buffer containing 1:500 Vybrant® DyeCycle^TM^ Green (V35005, Invitrogen) for 30 minutes at 37˚C. FACS was performed on a CytoFLEX S flow cytometer (Beckman Coulter) and analyses were conducted using CytExpert and FloJo software.

### Violet trace analysis

Cells were washed with 1x PBS and stained with 1x PBS containing 2 μM CellTrace^TM^ Violet (C34557, Invitrogen) for 7 minutes at 37˚C. 10 volumes ice-cold FBS were added and the cell mixture was centrifuged for 10 minutes at 250 g at 4˚C. Next, cells were washed twice with medium and plated. Two days later, cells were harvested and stained with Zombie NIR^TM^ and resuspended in FACS buffer for FACS analysis as described above.

### RNA extraction and purification for RNA sequencing and qPCRs

Cells were harvested in TRIzol^TM^ (Invitrogen), and chloroform extraction was performed. RNA was precipitated using 100% RNA-free ethanol to the aqueous phase and eluted in 100 uL MilliQ. Subsequently, the RNA was further purified using the NucleoSpin RNA kit (Macherey-Nagel) according to manufacturer's protocol.

### rRNA qRT-PCRs and analysis

Five micrograms of RNA were used for cDNA synthesis using random primers and the Super-Script^TM^ II Reverse Transcriptase kit (18064014, Invitrogen) according to the manufacturer's protocol. qRT-PCRs were performed on a C1000 thermal cycler (Bio-Rad) using SYBR green (170886, Bio-Rad) for the primers listed in S8 Table. For each primer set, triplicate reactions were performed and data analysis was performed by using the CFX Maestro Software (Bio-Rad). Expression levels were normalized to ATP5PO and UBE3B expression. Biological replicates were averaged and normalized by the wild-type average minus 1.

### iTHER patient data

For all iTHER patients informed written consent and ethical approval had been acquired before, as reported in Langenberg et al [48], including approval for future research. All data was pseudonymized and in the case of minor's consent was obtained from parents or legal guardians.

### RNA sequencing and analysis

For our isogenic model systems, we wanted to compare at least four mutant clones versus four wild-type clones. However, not for all isogenic models we acquired four clones (S1 Table) and therefore we split the cells into the required number of biological replicates. Before harvesting, we first maintained the splits of cells separately for at least 1.5 week, so that new independent mutations could be acquired.

For all isogenic clones, Illumina sequencing libraries were prepared using the Truseq RNA stranded RiboZeroPlus Kit (Illumina) and sequenced with 2x50bp paired-end sequencing on an Illumina Novaseq S1 (1500M) System. After sequencing the data was aligned to genome build GRCh37 (gencode v74) using STAR(v2.7.3a) and counts were generated using the R package Rsubread [49]. For all cell lines, Illumina sequencing libraries were prepared using the KAPA RNA HyperPrep Kit with RiboErase Kit (Roche) and sequenced with 2x150bp paired-

end sequencing on an Illumina NovaSeq6000 System. After sequencing, the reads were aligned to the genome (GRCh38; gencode v19) using STAR (version 2.7.0f) and counts were generated using the R package Rsubread [49]. For iTHER samples, Illumina sequencing libraries were prepared using the TruSeq RNA V2 Kit (Illumina) and sequenced with 2x100bp paired-end sequencing on an Illumina HiSeq4000 System. After sequencing the reads were aligned to the genome (GRCh37; gencode v17) using STAR(version 2.3.0e) and counts were generated using the R package Rsubread [49].

For all three datasets, counts were normalized by Variance Stabilizing Transformation (VST from DESeq2 package) and differentially expressed genes with an adjusted p-value less than 0.05 were determined using the DESeq2 [50] R package. For the GI-ME-N clones, we observed a batch effect in the wild-type clones. Therefore, we decided to compare the different GI-ME-N *ATRX* aberrant models only with their corresponding wild-type clones (generated by the same person). Gene ontology analysis was performed by inputting the overlapping differentially expressed genes in the gene ontology resource (http://geneontology.org/) for gene ontology biological process and Panther protein class [26] datasets using Fisher's exact test and FDR correction. GSEA was conducted using the R package fgsea with the gene ontology biological process and reactome gene set databases from MsigDB V7.2 [51]. Heatmaps were generated using the R package pheatmap [52], proportional Venn diagrams were created utilizing the eulerr [53, 54] package and the remaining figures were generated using the package ggplot2 [55].

## Supporting information

**S1 Fig. ALT-associated PML bodies (APBs) staining confirms the present of ALT in three patient-derived *ATRX* MED models.** White arrows mark co-localisation of telomeric (TelO) and PML foci, only a maximum of four arrows per image is displayed.
(PDF)

**S2 Fig. Southern blot confirms the present of ALT in three patient-derived *ATRX* MED models and violet trace identified unchanged proliferation rates.** (a) Southern blot containing one ALT negative cell line (SK-N-SH) and two well-known ALT positive neuroblastoma cell lines (SK-N-FI and LAN-6). All three PD$^{\Delta\Delta ATRX}$ models display long and heterogeneous telomeres and therefore confirm ALT. (b) Violet trace experiments on three *ATRX*$^{WT}$ and on three PD$^{\Delta\Delta ATRX}$ models.
(PDF)

**S3 Fig. Western blot confirmation of the generated *ATRX* aberrant isogenic models and absence of ALT.** (a) Western blots for ATRX protein of all the correct *ATRX* aberrant clones. Clones that were send for sequencing are marked with a red arrow on top. Green bar: confirmed wild-type clones, dark orange bar: confirmed *ATRX*$^{\Delta 2-10}$ clones or PD$^{\Delta\Delta ATRX}$ models, pink bar: confirmed *ATRX*$^{\Delta 2-13}$ clones and dark blue bar: confirmed *ATRX*$^{-/-}$ clones or the *ATRX*$^{-/-}$ osteosarcoma cell line U2OS. IFF: *ATRX* in-frame fusion protein product. Dark orange and pink asterisks indicate ATRX exon 2–10 and exon 2–13 MED IFF protein products, respectively. Stainings against α-tubulin were used as reference. (b) Southern blots confirming absence of long heterogeneous telomeres in our isogenic *ATRX* aberrant models. CHLA-90 and SK-N-MM were used as positive controls.
(PDF)

**S4 Fig. ALT-associated PML bodies (APBs) staining identifies absence of ALT in isogenic *ATRX* aberrant NB139 and SK-N-AS models.** White arrows mark co-localisation of telomeric (TelO) and PML foci, only a maximum of four arrows per image is displayed. CHLA-90

was added as positive control.
(PDF)

**S5 Fig. ALT-associated PML bodies (APBs) staining identifies absence of ALT in isogenic *ATRX* aberrant GI-ME-N$^{-/-}$ and GI-ME-N$^{\Delta 2-13}$ models.**
(PDF)

**S6 Fig. ALT-associated PML bodies (APBs) staining identifies absence of ALT in isogenic *ATRX* aberrant GI-ME-N$^{\Delta 2-10}$ models.**
(PDF)

**S7 Fig. Unaltered proliferation rates in the majority of the isogenic *ATRX* aberrant models.** (a-c) Violet trace experiments on *ATRX* wild-type and isogenic *ATRX* aberrant (a) GI-ME-N, (b) NB139 and (c) SK-N-AS models.
(PDF)

**S8 Fig. Principle component analysis (PCA) and Panther protein class analysis of generated isogenic *ATRX* aberrant models and PCA of patient-derived models and iTHER tumours.** (a) PCA of all generated isogenic *ATRX* aberrant clones showing separation based on the mother-lines. (b-f) PCA of (b) isogenic GI-ME-N clones, (c) isogenic NB139 clones, (d) isogenic SK-N-AS clones, (e) patient-derived *ATRX* aberrant and wild-type models (f) *ATRX* aberrant and wild-type iTHER tumours. (g-h) Significantly enriched Panther protein classes of the overlapping differentially expressed downregulated genes (g) for all *ATRX*$^{-/-}$ and *ATRX*$^{\Delta 2-13}$ isogenic models and (h) for all *ATRX*$^{\Delta 2-10}$ isogenic models.
(PDF)

**S9 Fig. The overlapping differentially expressed downregulated genes of all ATRX$^{\Delta 2-10}$ isogenic models are enriched for distinct RNA and metabolic processes according to GO BP.** Orange boxes highlighted the same terms as observed in Fig 3E.
(PDF)

**S10 Fig. Enrichment plots for all our performed GSEA for the GO BP gene set ribosome biogenesis.** Ribosome biogenesis enrichment plots for (a) GI-ME-N$^{\Delta 2-10}$ (b) NB139$^{\Delta 2-10}$ (c) PD$^{\Delta ATRX}$ (d) iTHER$^{\Delta 2-10}$ (e) GI-ME-N$^{-/-}$ (f) SK-N-AS$^{-/-}$ (g) GI-ME-N$^{\Delta 2-13}$.
(PDF)

**S11 Fig. Gene expression of the top 50 differentially expressed ribosome biogenesis genes confirms changes in ribosome biogenesis in *ATRX* isogenic and patient-derived models.** (a-c) Heatmaps showing expression values for the top 50 differentially expressed ribosome biogenesis genes that are normalized across all samples by Z-score in (a) SK-N-AS$^{-/-}$ (b) NB139$^{\Delta 2-10}$ and (c) PD$^{\Delta ATRX}$. Both row and column clustering were applied using the Euclidean distance.
(PDF)

**S12 Fig. Unchanged gene expression and protein expression in the majority of our isogenic models for several proteins that are involved in ribosome biogenesis.** (a) Western blot analysis for three proteins (MYCN, cMYC and EZH2) that are directly involved in regulating ribosome biogenesis revealed no changes that are consistent with our identified expression pattern dichotomy within *ATRX* aberrant models. Two distinct isoforms of MYCN are displayed in the right blot, with SK-N-AS being the only model expressing ΔMYCN. The color bars on top correspond to the *ATRX* status (green: wild-type, dark orange: exon 2–10 MED (or exon 2–9 in PD$^{\Delta ATRX}$ models), blue: knock-out, pink: exon 2–13 MED and yellow: exon 10–12 MED). Stainings against GAPDH were used as reference. (b) REST gene expression in patient-derived

*ATRX* MED and wild-type models. (c) REST gene expression in *ATRX* aberrant and wild-type iTHER tumours. (d) Western blot analysis showing unchanged *ATRX* IFF protein expression upon doxycycline induction of ATRX^WT protein expression. (e) Western blot analysis showing presence of HA-tagged ATRX^WT protein product only upon induction with doxycycline. (d-e) Stainings against α-tubulin were used as reference.
(PDF)

**S1 Table. Genetic overview of (un)successful clones validated by Sanger sequencing on both genomic DNA and RNA (cDNA).** RNA confirmed 'Yes' means absence of wildtype mRNA and for the MED models also detection of mRNA expression of the deleted allele. X-Chr: X-Chromosome.
(XLSX)

**S2 Table. The number of significantly downregulated and upregulated differentially expressed genes for all 8 analyses with an p-adjusted value of 0.05.**
(XLSX)

**S3 Table. All media and their components used for culturing the cell lines/organoids and isogenic clones.**
(XLSX)

**S4 Table. Sequences of sgRNAs used to make isogenic knock-out and ATRX MEDs.** The guide efficiency is shown as predicted by the CRISPOR design tool.
(XLSX)

**S5 Table. Primers used for cloning homology arm plasmids and PiggyBac doxycycline inducible *ATRX* wild-type plasmid.** Underlined sequences are homologous to the target sequence.
(XLSX)

**S6 Table. Primers used for validation of patient-derived and isogenic models.** Primers with a matching sequence are coloured with a similar colour and were used for validating different aberrations.
(XLSX)

**S7 Table. Antibodies used for western blotting and for immune fluorescence.**
(XLSX)

**S8 Table. Primers used for 47S rRNA qPCRs.**
(XLSX)

## Author Contributions

**Conceptualization:** Michael R. van Gerven, Jan Koster, Jan J. Molenaar, Marlinde L. van den Boogaard.

**Formal analysis:** Michael R. van Gerven, Linda Schild, Sander R. van Hooff.

**Funding acquisition:** Jan J. Molenaar.

**Investigation:** Michael R. van Gerven, Linda Schild, Jennemiek van Arkel, Bianca Koopmans, Luuk A. Broeils, Loes A. M. Meijs, Romy van Oosterhout, Marlinde L. van den Boogaard.

**Methodology:** Michael R. van Gerven, Marlinde L. van den Boogaard.

**Project administration:** Jan J. Molenaar.

**Resources:** Max M. van Noesel.

**Supervision:** Jan J. Molenaar, Marlinde L. van den Boogaard.

**Validation:** Michael R. van Gerven.

**Visualization:** Michael R. van Gerven, Marlinde L. van den Boogaard.

**Writing – original draft:** Michael R. van Gerven.

**Writing – review & editing:** Michael R. van Gerven, Luuk A. Broeils, Max M. van Noesel, Jan Koster, Sander R. van Hooff, Jan J. Molenaar, Marlinde L. van den Boogaard.

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
