## [Decision Letter · Decision Letter 0]

11 Jul 2023

Two opposing gene expression patterns within ATRX aberrant neuroblastoma

PONE-D-23-11824

Dear Dr. van den Boogaard,

We’re pleased to inform you that your manuscript has been judged scientifically suitable for publication and will be formally accepted for publication once it meets all outstanding technical requirements.

Kind regards,

Zhiming Li, Ph.D.

Academic Editor

PLOS ONE

Journal Requirements:

Reviewers' comments:

Reviewer's Responses to Questions

**Comments to the Author**

1. Is the manuscript technically sound, and do the data support the conclusions?

Reviewer #1: Yes

2. Has the statistical analysis been performed appropriately and rigorously? 

Reviewer #1: Yes

3. Have the authors made all data underlying the findings in their manuscript fully available?

Reviewer #1: Yes

4. Is the manuscript presented in an intelligible fashion and written in standard English?

Reviewer #1: Yes

5. Review Comments to the Author

Reviewer #1: Note this article has been reviewed by others and previous reviewers comments have been incorporated. This article describes interesting observations in a series of neuroblastoma cell lines which relate to understanding the role of ATRX in neuroblastoma oncogenesis. The introduction and discussion are releveant to the data presented.

6. PLOS authors have the option to publish the peer review history of their article (what does this mean?). If published, this will include your full peer review and any attached files.

Reviewer #1: No

---

## [Editor Report · Acceptance letter]

27 Jul 2023

PONE-D-23-11824 

Two opposing gene expression patterns within *ATRX* aberrant neuroblastoma 

Dear Dr. van den Boogaard:

I'm pleased to inform you that your manuscript has been deemed suitable for publication in PLOS ONE. Congratulations! Your manuscript is now with our production department. 

Kind regards, 

on behalf of

Dr. Zhiming Li 

Academic Editor

PLOS ONE